



# Is the Lorenz reference state global or local and observable?

Rémi Tailleux[1]

[1]Department of Meteorology, University of Reading, RG6 6ET Reading, United Kingdom

**Correspondence:** Rémi Tailleux (R.G.J.Tailleux@reading.ac.uk)

**Abstract.** Introduced over 70 years ago by Lorenz, the theory of available potential energy (APE) remains central to atmospheric and oceanic energetics. Yet the precise nature of its reference state is still debated and often misinterpreted. Because it is usually constructed from an energy-minimising adiabatic rearrangement of mass, the Lorenz reference state is commonly regarded as a global property of the fluid, requiring interactions between distant parcels. We argue instead that, analogously to the gravitational field, it should be viewed as a local and observable property of the *environment*. Gravity, though global in
origin, functions in practice as a local property measurable from its effect on falling bodies. Likewise, the Lorenz reference density and pressure profiles, $\rho_0(z)$ and $p_0(z)$, can be inferred from observations of buoyancy oscillations near equilibrium. Farther from equilibrium, a structural analysis of the governing equations shows that only deviations from the reference state affect motion, regardless of amplitude, thereby recovering Lorenz's APE separation as a structural property of fluid mechanics.
The Lorenz reference state is therefore best understood not as an arbitrary mathematical construct, but as an environmental constraint manifesting locally, reinforcing both the foundations of APE and its role in theories of ocean circulation and mixing.

## 1 Introduction

Despite its importance, the theoretical underpinnings of APE remain debated, especially regarding the physical nature of the
15 Lorenz reference state (LRS). In Lorenz's formulation, this state is defined as a minimum-energy configuration obtainable by an adiabatic rearrangement of mass (and, where relevant, salinity or moisture), yielding one-dimensional profiles of reference density $\rho_0(z)$ and pressure $p_0(z)$ (Huang, 2005; Saenz et al., 2015; Stewart et al., 2014). Because this construction relies on a global rearrangement, the LRS has often been interpreted as a global property of the fluid, seemingly implying long-range interactions between distant parcels and raising concerns about causality. This interpretation is conceptually prob-
20 lematic, however, because it makes it difficult to accept the validity of any explanation of local dynamical processes in terms of Lorenz-dependent quantities. For instance, it has recently been argued that the Lorenz reference density (LRD) surfaces (the density surfaces aligning with geopotential surfaces in the LRS) represent a physically more consistent definition of lateral stirring surfaces (Tailleux, 2016; Tailleux and Wolf, 2023; Tailleux, 2025) than the commonly accepted neutral surfaces (McDougall, 1987; Nycander, 2011; McDougall et al., 2014, 2017; Tailleux, 2017). However, if LRD surfaces were truly global in





character, their use would conflict with the assumed local nature of lateral stirring. We argue that this paradox is resolved if the LRS is not regarded as a property of the fluid itself, but rather as a property of the *environment* with which the fluid interacts. The fact that it is defined using fluid properties does not imply that it belongs to the fluid, any more than the gravitational field is a property of falling bodies. From this perspective, the global rearrangement is merely a formal device: what the dynamics 'feel' locally is the environmental constraint embodied by the reference state. This shift of view clarifies why the Lorenz refer-

ence state can legitimately define local stirring surfaces and helps reconcile its formal global construction with the requirement for locality in ocean mixing theory.

    This paper therefore addresses two interrelated questions: (i) can the LRS, while defined through a global construction, be regarded instead as a local and observable property of the environment? (ii) how does this environmental interpretation manifest itself in the structure of the governing equations of motion? We approach these questions from three complementary

perspectives: an analogy with gravity (Section 2); the observability of reference profiles through buoyancy oscillations and possible generalisations (Section 3); and a structural analysis of the governing equations that demonstrates their reliance on local deviations from the reference state (Section 4). A synthesis and outlook follow in Section 5.

## 2   Analogy with gravity

    To clarify the conceptual status of the LRS, it is instructive to draw an analogy with gravity. The gravitational acceleration

$g(x, y, z, t)$ is, in principle, set by the global distribution of mass in the Universe. More specifically, the value of $g$ at any location reflects the combined influence of distant bodies, from Earth's interior to the Sun and Moon. Yet in practice we treat $g$ as a locally measurable quantity, nearly constant at $g \approx 9.81 \, \text{m s}^{-2}$, without explicit reference to its global origin. This raises the central question: should we worry that gravity is global in character? Everyday physics shows that the answer is no, because local measurements suffice to determine $g$. For purely vertical motion, Newton's second law gives

$$45 \quad \frac{d^2 z}{dt^2} = -g, \tag{1}$$

which, for a free-fall experiment with initial conditions $z(0) = h$, $dz/dt(0) = 0$, leads to

$$g = \frac{2(h - z(t))}{t^2}. \tag{2}$$

Hence the value of $g$ can be inferred directly from the trajectory of a falling body. The dynamics of free fall depend only on the locally felt acceleration, not on the distant distribution of mass generating gravity. Were we unaware of Newton's law of

universal gravitation, there would be no reason to suspect that $g$ is determined by anything beyond local experiments.

    The *lesson* is that although gravity has a global physical origin, in practice it is indistinguishable from a local, observable property. Equally important, gravity is not a property of the falling body itself but of its environment, which constrains and governs the body's motion. This perspective offers a useful template for thinking about the Lorenz reference state: even if defined formally through a global rearrangement, its functional role in dynamics can in principle be inferred locally, without

reference to distant fluid parcels.





## 3 Observability of the Lorenz reference state

The analogy with gravity suggests that the key to resolving concerns over the LRS lies in the question of *observability*. If the reference profiles of density $\rho_0(z)$ and pressure $p_0(z)$ can be recovered from measurable properties of the fluid, then their practical status is local and physical, regardless of their formal global definition. Mathematically, the Lorenz reference state is

the property that allows one to predict the equilibrium depth $z_r(S,\theta)$ of a fluid parcel through the level of neutral buoyancy (LNB) equation (Tailleux, 2013a, 2018):

$$\rho(S,\theta,p_0(z_r)) = \rho_0(z_r). \tag{3}$$

Physically, this means that a parcel with salinity $S$ and potential temperature $\theta$, displaced adiabatically and isohalinely to its reference pressure $p_0(z_r)$, becomes neutrally buoyant relative to the reference profile. In this sense, the level $z_r$ embodies the

conceptual rearrangement central to defining the reference state.

*Crucially*, this abstract definition can, at least in principle, be tied to observable quantities. When the actual state is close to rest, parcels undergo adiabatic buoyancy oscillations about their equilibrium depths. The squared buoyancy frequency that governs these oscillations is predicted by

$$N_0^2(S,\theta,z) = -\frac{g}{\rho_0(z)}\left(\frac{d\rho_0}{dz} + \frac{g\,\rho_0(z)}{c_{sb}^2}\right), \tag{4}$$

(Tailleux, 2013b), where $c_{sb}$ is the speed of sound. In principle, measurements of buoyancy oscillations at multiple depths can be used to infer $d\rho_0/dz$ and, by vertical integration, reconstruct the reference profiles $\rho_0(z)$ and $p_0(z)$. The constants of integration may then be constrained by knowledge of the total mass of the fluid and the surface pressure.

For small departures from rest, this procedure renders $\rho_0(z)$ and $p_0(z)$ effectively *measurable* through local experiments, much like free-fall experiments determine the local value of $g$. Importantly, what such experiments reveal is not a hidden

property of the parcels themselves, but the environmental constraint to which they must adjust, in line with the action–reaction principle. For larger departures, buoyancy oscillations are modified by background flows and nonlinear effects, so the simple relation (4) no longer suffices. Still, we expect the LRS to remain tied to observable dynamical behaviour, although extracting properties from more complex motions might be impractical. Hence, even though the Lorenz reference state is defined formally through a global rearrangement, it emerges in practice as an empirically accessible environmental property. This empirical

accessibility is mirrored in the governing equations, where the LRS emerges as a structural constraint on local deviations, as developed in next Section 4.

## 4 Structural basis for the Lorenz reference state

Beyond analogies and observability arguments, the relevance of the LRS can also be demonstrated directly from the structure of the governing equations. Recently, Tailleux and Dubos (2024) introduced a new framework, called *Static Energy Asymptotics*

(SEA), allowing one to write the full Navier-Stokes equations and most standard sound-proof approximations of them by means of a universal formulation. In their framework, the momentum and continuity equations of a stratified, rotating fluid may be





expressed in terms of the static energy function $\Sigma = \Sigma(\eta, S, p, \Phi)$, which depends on specific entropy $\eta$, composition $S$ (e.g. salinity), pressure $p$, and geopotential $\Phi = gz$:

$$\frac{D\mathbf{v}}{Dt} + 2\mathbf{\Omega} \times \mathbf{v} + \frac{\partial \Sigma}{\partial p}\nabla p + \frac{\partial \Sigma}{\partial \Phi}\nabla\Phi = \mathbf{F}, \tag{5}$$

$$\nabla \cdot \mathbf{v} = \frac{D}{Dt}\ln\frac{\partial \Sigma}{\partial p}, \tag{6}$$

where $\mathbf{v} = (u, v, w)$ is the three-dimensional velocity vector, $\mathbf{\Omega}$ the Earth's rotation vector, and $\mathbf{F}$ the viscous force. A key observation is that $\Sigma$ influences motion only through its derivatives with respect to $p$ and $\Phi$. This makes it possible to decompose the static energy into dynamically active and passive contributions:

$\Sigma = \Sigma_{\mathrm{dyn}}(\eta, S, p, \Phi) + \Sigma_{\mathrm{heat}}(\eta, S),$ \hfill (7)

where $\Sigma_{\mathrm{dyn}}$ controls the evolution of the system, while $\Sigma_{\mathrm{heat}}$ is invisible to the dynamics. Lorenz's original insight can therefore be reinterpreted structurally: the passive part $\Sigma_{\mathrm{heat}}$ is simply the value of $\Sigma$ evaluated at the reference state, that is,

$$\Sigma_{\mathrm{heat}}(\eta, S) = \Sigma(\eta, S, p_0(\Phi_r), \Phi_r), \tag{8}$$

with the reference position $\Phi_r = gz_r$ defined by the level of neutral buoyancy condition $\rho(\eta, S, p_0(\Phi_r)) = \rho_0(\Phi_r)$ (same as Eq. (3) but defined in terms of geopotential $\Phi$ rather than $z$). By substituting $\Sigma \to \Sigma_{\mathrm{dyn}}$ in Eqs. (5)–(6), and after some simple algebra, the momentum equation can be recast in a form that makes the influence of the Lorenz reference state on local dynamics explicit:

$$\frac{D\mathbf{v}}{Dt} + 2\mathbf{\Omega} \times \mathbf{v} + \nabla\Sigma_{\mathrm{dyn}} = \underbrace{\frac{\partial \Sigma_{\mathrm{dyn}}}{\partial \eta}\nabla\eta + \frac{\partial \Sigma_{\mathrm{dyn}}}{\partial S}\nabla S}_{\mathbf{P}_a} + \mathbf{F}. \tag{9}$$

All quantities linked to $\Sigma_{\mathrm{dyn}}$ are proportional to $(\Phi - \Phi_r)$, i.e. to the distance from equilibrium. Tailleux and Wolf (2023) showed that, in most of the ocean interior outside the Southern Ocean, the thermodynamic buoyancy force vector $\mathbf{P}_a$ is nearly parallel to the standard neutral vector that has long been regarded as fundamental for describing lateral stirring (McDougall, 1987; Jackett and McDougall, 1997). For steady, inviscid, adiabatic, and isohaline flow, $\mathbf{v} \cdot \nabla\Sigma_{\mathrm{dyn}} = 0$ and $\mathbf{P}_a \cdot \mathbf{v} = 0$, which may be exploited to derive an explicit expression for the absolute velocity field (Tailleux, 2023).

Finally, to connect this approach with APE theory, recall that potential energy $E_p$ may be written in terms of $\Sigma$ as

$$E_p = \Sigma - p\frac{\partial \Sigma}{\partial p}, \tag{10}$$

(Tailleux and Dubos, 2024) with the available and non-available parts given by

$$E_a = \Sigma_{\mathrm{dyn}} - p\frac{\partial \Sigma_{\mathrm{dyn}}}{\partial p}, \qquad E_b = \Sigma_{\mathrm{heat}}. \tag{11}$$





Thus, the separation into available and unavailable energy components emerges naturally from the governing equations themselves, independent of analogy or interpretive construction. This establishes that APE theory is not only a powerful diagnostic framework but also a direct structural consequence of fluid dynamics. Equally importantly, it shows that the Lorenz reference state enters the equations in the way an *external constraint* would: it shapes the dynamics through its deviations, but does not need to be regarded as an intrinsic property of the parcels themselves. This perspective connects the structural view back to the earlier analogy and observability arguments, and further clarifies why Lorenz reference density surfaces can legitimately define local stirring directions.

## 5  Discussion and conclusions

Because it is generally defined in terms of a global rearrangement of parcels, the LRS is often thought to represent a global property of the fluid requiring long-range interactions between distant parcels. From examining the issue from three complementary perspectives, we conclude that it is better understood instead as a local property of the environment with which the fluid interacts. Our argument builds on the analogy with gravity, which shows that quantities with a global physical origin can nevertheless function as locally observable environmental properties. Gravitational acceleration $g$ is determined by the overall mass distribution of the Universe, yet it is only felt locally by a falling body and can therefore be estimated directly from simple experiments. Likewise, the LRS may require a formal global construction, but dynamically it manifests as an environmental constraint accessible to local observations. Near equilibrium, the LRS controls the frequency of buoyancy oscillations, whose measurements can in principle be inverted to reconstruct the vertical gradients of the reference density and pressure profiles $\rho_0(z)$ and $p_0(z)$. This shows that the reference state is not merely a mathematical abstraction, but a physical constraint that the environment imposes on the fluid, consistent with the action–reaction principle. Far from equilibrium, a structural analysis of the governing equations confirms that only deviations from the reference state affect motion. By decomposing the static energy into dynamical and passive components, one recovers Lorenz's separation of total potential energy into available and unavailable parts as a direct structural property of the Navier–Stokes equations. This reinforces the idea that the reference state is built into the framework of the dynamics themselves, but as an external constraint rather than an internal property of the fluid.

Taken together, these perspectives establish that the LRS, although defined formally by a global rearrangement, functions dynamically as a local and observable environmental property. The profiles $\rho_0(z)$ and $p_0(z)$ influence motion in the same way that gravity constrains free fall: their ultimate causes are global, but their dynamical role is local. Recognising this resolves objections about non-locality and causality, and strengthens the case for Lorenz reference density (LRD) surfaces as a more physically consistent definition of lateral stirring than neutral surfaces (Tailleux, 2016; Tailleux and Wolf, 2023; Tailleux, 2025). Future progress will depend on clarifying whether alternative formulations of the reference state may also be valid, and on developing observational strategies capable of diagnosing the most appropriate environmental constraint in practice (see, e.g., Wong et al. (2016); Harris and Tailleux (2025)). *In conclusion*, the essential point is that the Lorenz reference state should be understood as an environmental property that the fluid responds to locally, just as a falling body responds to the local value



of gravity. This interpretation strengthens the theoretical foundations of APE and underpins its application to modern problems of ocean circulation, energetics, and mixing.

*Author contributions.* The author designed the research and wrote the paper.

*Competing interests.* The author declares no competing interests

*Acknowledgements.* The idea for this paper grew out of many discussions with colleagues and debates with reviewers over the years. The author acknowledges useful discussions with Geoff Stanley, Stephen Griffies, Brian Hoskins, Thomas Dubos, Guillaume Roullet, and Ben Hatton. This research has been supported by the NERC-funded OUTCROP project (grant no. NE/R010536/1).



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
