# Peer review of "Is the Lorenz reference state global or local and observable?"

_EGUsphere, 2025_

## Referee Comment (RC1)

Discussion (started: 2 October 2025) of the preprint at egusphere:

Is the Lorenz reference state global or local and observable?

By Rémi Tailleux, <a href="https://doi.org/10.5194/egusphere-2025-4595">https://doi.org/10.5194/egusphere-2025-4595</a>

**Discussion:**

Estuaries with a pronounced, slanted halocline between discharged brackish river water at the surface, and ocean water intruding along the sea floor in opposite direction, represent typical oceanographic examples for substantial amounts of available potential energy (APE) stored in the water column. Such systems may offer a very transparent demonstration of the advantageous approach suggested by Rémi Tailleux's proposal. In such cases, the distinct water masses above and below the halocline may each be assumed for simplicity to be well-mixed so that high-energy buoyancy oscillations with high frequencies are restricted to the halocline itself. Then, the related APE may easily be estimated quantitatively from available local CTD profiles, assumingly in total being proportional to the lateral surface area of the halocline. By contrast, globally the APE is expected to be proportional to the pycnocline slope and the related up- and downwelled water volumes above and below.

To assist readers in taking advantage, the author should explicitly demonstrate his new method at a simple analytical tutorial example, perhaps in an appendix of the paper.

The Baltic Sea is a special marine system that is extremely well studied by dense spatial and temporal oceanographic long-term monitoring (Feistel et al. 2008). Its typical estuarine circulation (Reissmann et al. 2009, Burchard et al. 2018) is mainly driven by strong lateral salinity gradients (Feistel et al. 2010) and the related Available Potential Energy (APE). The recent paper of Rémi Tailleux, as I understand it, suggests an elegant method of regularly estimating the Baltic APE from routine CTD profiles in combination with the TEOS-10 seawater standard (IOC et al. 2010) for computing the local vertical stability with respect to fluctuations of pressure at constant salinity S and entropy  $\eta$ , as

$$\left[ \frac{\mathrm{d}\rho}{\mathrm{d}p} - \left( \frac{\partial\rho}{\partial p} \right)_{S,\eta} \right] = \left[ \left( \frac{\partial\rho}{\partial s} \right)_{p,\eta} \frac{\mathrm{d}s}{\mathrm{d}p} + \left( \frac{\partial\rho}{\partial\eta} \right)_{S,p} \frac{\mathrm{d}\eta}{\mathrm{d}p} \right] \text{, see the paper's eq. (4)}.$$

The Baltic Sea may provide such a simple idealized system for tutorial purpose. For simplicity, let the Baltic basin itself be represented by a box homogenously filled with brackish water, and the North Sea by a similar box with salty ocean water, as shown in Fig. 1. The natural shallow connection between the two, given by the Danish Belt Sea, may be modelled by narrow openings permitting gentle exchange between the basins in both directions, which in nature is enforced by tidal oscillations (Feistel et al. 2004) or wind-driven anomalies of the sea level (Matthäus 2006).

Of the schematic model in Fig. 1, the state of minimum potential energy is easily constructed by global displacement of the water parcels as shown in Fig. 2. Note that the situation of Fig. 1 may be arbitrarily close to rest as the salinity difference between the basins may be imagined as small as required.

Comparing Figs. 1 and 2, the available potential energy results from the difference between a vast volume of dense water transferred from the North to the Baltic Sea and lowered to the bottom there, and an equal volume of less dense water in opposite direction raised to the surface. In reality, this potential energy drives the natural estuarine circulation of the Baltic. It is unclear, however, how this amount of energy may be computed from the size, shape and local steepness of the halocline in

the shallow transition zone of Fig. 1. Moreover, the halocline in Fig. 2 is much more extended than before, while there is no APE associated anymore with the reference state of Fig. 2, by definition. Clarification is desired.

Fig. 1 Simplified schematic of the Baltic Sea as a basin with brackish water, on the right-hand side, and the North Sea as a basin with salty ocean water, on the left. Haloclines separating the two water masses are located in narrow connections with volumes negligible in comparison to the basins.

Fig. 2 Theoretical state of minimum potential energy obtained from Fig. 1 by global rearrangement of the water parcels. The APE of Fig. 1 is easily estimated from the differences between the two figures. Although Fig. 2 has zero APE, the pycnocline is still existing and is even larger than in Fig. 1. Buoyancy oscillations are possible only at that pycnocline because elsewhere the water bodies are assumed to be neutrally stratified with vanishing vertical gradients of salinity  $\frac{\mathrm{d}s}{\mathrm{d}p}=0$  and entropy  $\frac{\mathrm{d}\eta}{\mathrm{d}p}=0$  (turbulently mixed with isentropic lapserate, McDougall and Feistel 2003).

If it possibly turns out that the proposed Tailleux method of estimating the APE from local CTD profiling is not applicable to cases of an estuarine like the Baltic, the related validity limits should be explicitly addressed in the article.

**Minor issues:**

- The term "equilibrium" in the abstract and elsewhere should be more specific: is it thermodynamic, hydrodynamic or mechanic equilibrium, for example?
- The "static energy function"  $\Sigma$  should be specified in terms of proper conventional thermodynamic energies of TEOS-10, such as enthalpy h or Helmholtz energy f, etc. How is  $\Sigma$  to be computed from the TEOS-10 software library?
- How is  $\Sigma_{heat}$  related to potential enthalpy (McDougall et al. 2021) ?
- In eq. (5) and below, the partial derivatives should be written in thermodynamic convention, indicating which variables are kept constant.

- Eqs. (10) and (11) are typical thermodynamic Legendre transforms (Alberty 2002). Presenting exact differentials of  $\Sigma$ ,  $\Sigma_{\rm dyn}$ ,  $\Sigma_{\rm heat}$ ,  $E_p$ ,  $E_a$ ,  $E_b$  along with explaining the physical quantities of the implied partial derivatives in terms of standard textbook thermodynamics would greatly help the reader to follow the given arguments.

**Conclusion:**

It is recommended that the submitted paper will be subject to a **MAJOR REVISION** before it may be published.

**References**

Alberty, R.A. (2002): Use of Legendre transforms in chemical thermodynamics. International Union of Pure and Applied Chemistry, Physical Chemistry Division, Commission on Thermodynamics. The Journal of Chemical Thermodynamics 34, 1787-1823. <a href="https://doi.org/10.1016/S0021-9614(02)00170-2">https://doi.org/10.1016/S0021-9614(02)00170-2</a>

Burchard, H., Bolding, K., Feistel, R., Gräwe, U., Klingbeil, K., MacCready, P., Mohrholz, V., Umlauf, L., van der Lee, E. (2018): The Knudsen theorem and the Total Exchange Flow analysis framework applied to the Baltic Sea. Progress in Oceanography 165, 268-286, https://doi.org/10.1016/j.pocean.2018.04.004

Feistel, R., Nausch, G., Matthäus, W., Lysiak-Pastuszak, E., Seifert, T., Sehested Hansen, I., Mohrholz, V., Krüger, S., Buch, E., Hagen, E. (2004): Background Data to the Exceptionally Warm Inflow into the Baltic Sea in Late Summer of 2002. Meereswissenschaftliche Berichte Warnemünde 58, 1-58, <a href="https://www.io-warnemuende.de/files/forschung/meereswissenschaftliche-berichte/mebe58">https://www.io-warnemuende.de/files/forschung/meereswissenschaftliche-berichte/mebe58</a> 2004 paper.pdf

Feistel, R., Nausch, G., Wasmund, N. (Eds., 2008): State and Evolution of the Baltic Sea, 1952 – 2005. A Detailed 50-Year Survey of Meteorology and Climate, Physics, Chemistry, Biology, and Marine Environment. John Wiley & Sons, Inc., Hoboken, 704 pp., ISBN 978-0-471-97968-5

Feistel, R., Weinreben, S., Wolf, H., Seitz, S., Spitzer, P., Adel, B., Nausch, G., Schneider, B., Wright, D.G. (2010): Density and Absolute Salinity of the Baltic Sea 2006-2009. Ocean Science 6, 3-24, https://doi.org/10.5194/os-6-3-2010

IOC, SCOR, IAPSO (2010): The international thermodynamic equation of seawater - 2010: Calculation and use of thermodynamic properties. Written by: McDougall, T.J., Feistel, R., Wright, D.G., Pawlowicz, R., Millero, F.J., Jackett, D.R., King, B.A., Marion, G.M., Seitz, S., Spitzer, P., Chen, C.T.A. Intergovernmental Oceanographic Commission, Manuals and Guides No. 56, UNESCO (English), Paris, 196 pp., ISBN 978-7-5027-8151-4, http://www.teos-10.org

Matthäus, W. (2006): The history of investigation of salt water inflows into the Baltic Sea - from the early beginning to recent results. <a href="https://www.io-warnemuende.de/files/forschung/meereswissenschaftliche-berichte/mebe65">https://www.io-warnemuende.de/files/forschung/meereswissenschaftliche-berichte/mebe65</a> 2006.pdf

McDougall, T.J., Feistel, R. (2003): What Causes the Adiabatic Lapse Rate? Deep-Sea Research 50, 1523-1535, https://doi.org/10.1016/j.dsr.2003.09.007

McDougall, T.J., Barker, P.M., Holmes, R.M., Pawlowicz, R. Griffies, S.M., Durack, P.J. (2021): The interpretation of temperature and salinity variables in numerical ocean model output and the

calculation of heat fluxes and heat content. Geosci. Model Dev. 14, 6445–6466, <a href="https://doi.org/10.5194/gmd-14-6445-2021">https://doi.org/10.5194/gmd-14-6445-2021</a>

Reissmann, J.H., Burchard, H., Feistel, R., Hagen, E., Lass, H.U., Mohrholz, V., Nausch, G., Umlauf, L., Wieczorek, G. (2009): State-of-the-art review on vertical mixing in the Baltic Sea and consequences for eutrophication. Progress in Oceanography 82, 47–80, <a href="https://doi.org/10.1016/j.pocean.2007.10.004">https://doi.org/10.1016/j.pocean.2007.10.004</a>

---

## Referee Comment (RC3)

Comment on "Is the Lorenz reference state global or local and observable?"

The manuscript addresses an important conceptual question in ocean energetics: whether the Lorenz reference state (LRS) should be understood as a global or local quantity, and to what extent it is observable. The LRS, defined as the adiabatic rearrangement of water parcels minimizing potential energy, has become a fundamental construct in diagnosing available potential energy. However, its physical interpretation remains debated, as it depends on the global distribution of entropy and salinity, while the governing equations of motion are local. The author aims to reconcile this apparent inconsistency through analogies and illustrative examples.

Overall, I found the manuscript stimulating and appreciated several original and thought-provoking insights. Nevertheless, I am unable to recommend publication in its current form, for the reasons outlined below.

1. On the use of gravitational acceleration as an analogy. The manuscript treats gravitational acceleration as an example of a field with a global origin that is locally observable. However, this analogy is problematic. The present formulation implies that gravitational acceleration can be described as a function of space and time related to a scalar potential,  $g = |\nabla \Phi|$ , with  $\Phi(x, y, z, t)$  determined instantaneously by the mass distribution of the universe via Newton's law. This view contradicts the modern understanding of gravity in general relativity.

In general relativity, gravitational acceleration is frame-dependent; one can always find a freely falling frame in which gravity vanishes locally. A familiar example is the International Space Station (ISS): although Earth's gravitational attraction at the ISS is nearly the same as at the surface, astronauts experience weightlessness because their orbital motion balances the gravitational pull with centrifugal acceleration. Hence, the concept of a universally measurable g is not meaningful in a local experiment.

More fundamentally, gravity is a manifestation of spacetime geometry. The gravitational field is expressed through the metric tensor and propagates at the finite speed of light, analogous to waves in an elastic medium. Thus, although gravity originates from the distribution of mass and energy in the universe, it is best regarded as a transient local property rather than a static global one.

I understand that the author introduces gravity as a pedagogical analogy to frame the subsequent discussion of ocean energetics. Nevertheless, even for illustrative purposes, the description should remain scientifically accurate. While such an analogy might be acceptable in an informal essay, a refereed article requires stricter adherence to established physical theory.

- 2. On the observability of the Lorenz reference state. The manuscript suggests that the LRS can, in principle, be reconstructed from locally measured buoyancy frequency if the system is sufficiently close to rest. However, this assumption of near-equilibrium conditions is rarely met in the real ocean. In particular, the pronounced tilting of isopycnal surfaces in high-latitude regions indicates significant departures from equilibrium. Consequently, the practical and theoretical usefulness of the arguments presented in Section 3 is unclear.
- 3. On the formulation using the static energy function. Section 4 presents a reformulation of the Navier–Stokes equations in terms of a static energy function, following the

author's earlier work (Tailleux and Dubos, 2024). This framework is indeed elegant and potentially valuable for numerical modeling. However, the manuscript's interpretation appears overstated.

As written in equations (5)–(7), the equations of motion are expressed using a static energy function  $\Sigma$ , decomposed into a dynamically relevant component  $\Sigma_{\text{dyn}}(\eta, S, p, \Phi)$  and a passive component  $\Sigma_{\text{heat}}(\eta, S)$ . The author defines  $\Sigma_{\text{heat}}$  based on the LRS and argues that the LRS "enters the equations in the way an external constraint would." Yet, this statement is questionable, as the choice of  $\Sigma_{\text{heat}}$  in equation (8) is not unique. One could equally adopt another reference state  $z'_r(\eta, S)$  without altering the subsequent formulation. This arbitrariness weakens the claim that the LRS directly shapes the dynamics through its deviations. Although the LRS remains a useful diagnostic construct, its role as a physically necessary ingredient of the governing equations is not convincingly demonstrated.

In summary, while the manuscript raises important conceptual questions and offers stimulating reflections, the arguments are not fully justified by the proposed analogies and theoretical formulations. Despite these concerns, I found the paper intellectually engaging and potentially valuable as a discussion piece. I would encourage the author to make it publicly available as a non-refereed contribution, as it may inspire future theoretical advances in the energetics of the general ocean circulation.

---

## Referee Comment (RC4)

**Response to the Author**

I appreciate the author's efforts in addressing the reviewers' comments. However, after carefully evaluating the responses, I find that several major concerns remain insufficiently resolved.

1. On the use of gravitational acceleration as an analogy. Even when restricting the discussion to Newtonian mechanics, gravity does not seem to constitute an appropriate analogy for the Lorenz reference state (LRS). The gravitational acceleration g is related to a scalar potential field  $\Phi$  through  $g = |\nabla \Phi|$ , and this potential satisfies

$$\nabla^2 \Phi = 4\pi G \rho,\tag{1}$$

where G is the gravitational constant and  $\rho(x, y, z, t)$  is the mass distribution. In this sense, the gravitational field can be described by a Poisson equation, which is formally local in space. In the oceanographic context, a possible explicit analogue of  $\Phi$  would be the pressure p. Although the pressure force in an incompressible flow acts nonlocally, a similar Poisson equation for p can be written to illustrate how hydrodynamic motion at a given point influences adjacent fluid elements. These processes are conceptually straightforward.

The LRS, by contrast, is a far more intricate construct. Determining the LRS corresponding to a given spatial distribution of S and  $\theta$  requires a global rearrangement of fluid parcels, which is inherently difficult to express in a local form. This is precisely why the physical interpretation of the LRS has remained a challenging issue.

2. On the observability of the Lorenz reference state. The theoretical argument presented through equations (2)–(5) is not convincing. While the mathematical manipulations themselves are correct, the reference profiles  $\rho_0(z)$  and  $p_0(z)$  introduced here may be chosen arbitrarily; there is no inherent reason to identify them with the LRS.

Furthermore, the statement that "while  $p_0(z)$  enters as a passive reference, its choice determines  $N_0$  and is thus constrained by observations" is not meaningful. The authors may be implicitly assuming that  $N_0$  corresponds to the in-situ Brunt-Väisälä frequency, but this is not the case. The true Brunt-Väisälä frequency is determined by the vertical gradients of salinity and entropy. In the present notation, it is given by

$$N^2 = b_S(S, \theta, z) S_z + b_\theta(S, \theta, z) \theta_z.$$
(2)

In contrast, the quantity  $N_0$  defined in the author's reply depends solely on the arbitrarily chosen reference state and is therefore not observable.

3. On the formulation using the static energy function. As I noted in my previous comments, the decomposition of the static energy function into  $\Sigma = \Sigma_{\rm dyn} + \Sigma_{\rm heat}$  and the use of the LRS to define the latter is an interesting and potentially useful idea. However, I remain unconvinced by the strengthened arguments presented in the author's response.

The author now discusses a specific example involving a diabatic process at the ocean surface. In this situation, defining  $\Sigma_{\text{heat}}$  based on the LRS makes expression (6) represent the APE production rate. While this observation is valid, it is not unexpected, since APE is fundamentally defined relative to the LRS. This simple example thus reiterates a well-known result: destabilization of the stratification and the onset of convection are linked to the APE budget. It does not, however, substantively support the central claim of the manuscript that "the Lorenz reference state enters the equations in the way an external constraint would."

---

## Author Comment (AC1)

**Reply to Rainer Feistel**

Many thanks to Prof. Feistel for useful comments, which we anticipate will help clarify the arguments made in my paper and hopefully help the reader recognise their importance.

**Major points**

**Comment.** To assist readers in taking advantage, the author should explicitly demonstrate his new method at a simple analytical tutorial example, perhaps in an appendix of the paper.

**Response.** We thank Prof. Feistel for the constructive suggestion. Before addressing the tutorial aspect, we clarify a central point where, we believe, a misunderstanding arises.

First, the quantity appearing in our Eq. (4),

$$N_0^2(S, \theta, z) = -\frac{g}{\rho_0(z)} \left( \frac{d\rho_0}{dz} + \frac{g \,\rho_0(z)}{c_s^2} \right),\tag{1}$$

is defined from the *environmental* reference profiles  $(\rho_0(z), p_0(z))$ , and of the square speed of sound  $c_s^2$  of the fluid parcel. Hence, it is an extrinsic function of state, that is, a joint property of the environment and of the fluid parcel. By contrast, the locally defined squared buoyancy frequency used by Prof. Feistel is based on the *local* parcel derivatives,

$$\left[ \frac{d\rho}{dp} - \frac{\partial\rho}{\partial p} \Big|_{S,\eta} \right] = \left[ \frac{\partial\rho}{\partial S} \Big|_{\eta,p} \frac{dS}{dp} + \frac{\partial\rho}{\partial\theta} \Big|_{S,p} \frac{d\theta}{dp} \right],$$
(2)

where  $d\rho/dp$  represents the vertical gradient of the local in-situ density profile. His expression is therefore exclusively a property of the water colum and hence of the fluid only. When the actual state is close to its mechanically balanced resting state, (1) and (2) can be approximately consistent. However, in the Baltic Sea toy example invoked by Prof. Feistel, the flow is far from rest, so  $N_0^2$  from (1) generally differs substantially from the local buoyancy frequency computed from CTD profiles via (2). This limitation is explicitly acknowledged in the paper (lines 76–78): "For larger departures, buoyancy oscillations are modified by background flows and nonlinear effects, so the simple relation (4) no longer suffices. Still, we expect the LRS to remain tied to observable dynamical behaviour, although extracting properties from more complex motions might be impractical."

Context and scope. The construction of Lorenz-type global APE from a local principle—hence a non-negative APE density computable for each fluid parcel from CTD or climatology—has been established for decades [1, 2, 3]. Its practical utility has been demonstrated across diverse problems: the ocean

energy cycle [4], atmospheric storm tracks [5, 6], tropical cyclone intensification [7, 8], turbulent stratified mixing [9, 10, 11], double-diffusive instabilities [12], and even magneto-thermal turbulence in astrophysics [13].

Applications to semi-enclosed or marginal basins (e.g. the Baltic) are of high interest but are *outside the scope* of the present paper. A fundamental open question is whether such basins "feel" the same environment as the global ocean; that is, whether the same  $(\rho_0(z), p_0(z))$  can meaningfully serve both the World Ocean and the Baltic Sea in defining APE. This issue has been touched on [14, 11] but remains unresolved. The present paper focuses on interpretative aspects of local APE where the use of *single* reference profiles is appropriate (implicitly, sufficiently simple domains).

Main contributions. (i) We provide an argument that, in local APE theory, the Lorenz reference state (LRS) is best interpreted as a property of the *environment*, not of the fluid parcel. (ii) We give a new proof that the APE/BPE partition, locally and globally, is a *structural* property of the Navier–Stokes equations.

**Comment.** If it turns out that the proposed method of estimating APE from local CTD profiling is not applicable to estuarine cases like the Baltic, the related validity limits should be explicitly addressed.

Response. We agree that validity limits should be clearly stated. Importantly, the structural result established in Section 4 (building on [15]) shows that the partition of potential energy into dynamically active and passive components is an exact property of the Navier–Stokes equations. In this sense, local APE theory is always applicable, at least in principle. What remains nontrivial in practice is the choice of environmental reference state, which determines  $\Sigma_{\rm heat}$  and the APE definition itself. Theory predicts that an optimal partition  $\Sigma_{\rm heat}/\Sigma_{\rm dyn}$  must exist, but it does not yet uniquely prescribe it; hence heuristics are sometimes needed.

If one accepts that the LRS is an environmental property, then a definitive formulation of APE in complex settings awaits a satisfactory theory of the "environment." Any perceived limitation in practice thus lies not with APE as a concept (which has rigorous foundations) but with our current, incomplete understanding of how to define the environment and the equations governing it.

**Illustration.** This sensitivity is well illustrated by [8] in the context of tropical cyclone (TC) intensification. The volume-integrated APE budget can be written

$$\frac{dAPE}{dt} = G_A - C(APE, KE), \tag{3}$$

where  $G_A$  is the APE production by surface enthalpy fluxes and C(APE, KE) is the APE-to-KE conversion. In (3), APE, dAPE/dt, and  $G_A$  all depend on the reference state, while C(APE, KE) does not. Figure 1 shows that one can select a reference state for which  $G_A$  closely tracks the reference-state-independent

Figure 1: Comparison of APE production rates  $G_A$  for different reference states (colours) versus the APE-to-KE conversion C(APE, KE) (black). See [8] for details and assumptions.

conversion C(APE,KE), thereby identifying a "best" environment for that problem.

Implications for the Baltic. By the same logic, there must exist an appropriate environmental specification for computing APE density in the Baltic. The fact that a universally accepted choice is not yet established reflects an open theoretical question about the definition of environment in semi-enclosed basins, not a failure of APE theory. This challenge is not unique to APE: the notion of environment is also central in thermodynamics (e.g. practical exergy calculations). Its resolution will likely have implications beyond APE.

On a tutorial example. We are open to the idea of adding some kind of illustrative examples that could help the reader better understand our arguments, and will try to do so in the revision provided that this is feasible within space constraints.

**Minor points**

The term "equilibrium" in the abstract and elsewhere should be more specific: is it thermodynamic, hydrodynamic or mechanic equilibrium, for example. The term "equilibrium" corresponds to that underlying theory of available potential energy, that is, a notional rest state obtainable from the actual state by means of an adiabatic and isohaline re-arrangement

of mass. This corresponds to a mechanical equilibrium. We'll make sure to be more specific when revising the paper.

The "static energy function"  $\Sigma$  should be specified in terms of proper conventional thermodynamic energies of TEOS-10, such as enthalpy h or Helmoltz energy f, etc. How is  $\Sigma$  to be computed from the TEOS-10 software library?

Static energy  $\Sigma$  is defined as the sum of specific enthalpy and geopotential,

$$\Sigma(\eta, S, p, \Phi) = h(\eta, S, p) + \Phi = h(\eta, S, p) + gz, \tag{4}$$

so that its 'natural' canonical dependent variables are  $(\eta,S,p,\Phi)$ . If the pressure can be approximated as hydrostatic so that  $dp/d\Phi\approx-\rho$ , its evolution equation can be approximated as

$$\frac{D\Sigma}{Dt} = T \frac{D\eta}{Dt} + \mu \frac{DS}{Dt} + \frac{1}{\rho} \frac{Dp}{Dt} + \frac{D\Phi}{Dt}
\approx -\frac{1}{\rho} \nabla \cdot \mathbf{J}_h + \varepsilon_K + \frac{1}{\rho} \frac{D_h p}{Dt},$$
(5)

where  $D_h/Dt = \partial/\partial t + \mathbf{u} \cdot \nabla_h$  only contains horizontal advection by the ageostrophic component of the velocity field. This is very similar to the evolution equation for specific enthalpy

$$\rho \frac{Dh}{Dt} = -\nabla \cdot \mathbf{J}_h + \rho \varepsilon_K + \frac{Dp}{Dt} \tag{6}$$

As a result,  $\Sigma$  is much less affected by compressibility effects than specific enthalpy. Here, non-conservative effects are the sum of viscous dissipation rate  $\varepsilon_K$  and of the  $D_h p/Dt$ . Based on observations [16],  $\varepsilon_K$  tends to like within  $10^{-10}Wkg^{-1}$  and  $10^{-8}Wkg^{-1}$ . As regards to  $D_h p/Dt$ , [17] estimated the vertical integral of  $\mathbf{u} \cdot \nabla_h p$  in two different coupled climate models. They found that it rarely exceed  $10mWm^{-2}$ , which averaged over the mean depth of the ocean 1000m corresponds to an energy conversion rate of about  $10^2mWm^3$  or  $10^{-8}Wkg^{-1}$ , similar to the upper range of observed viscous dissipation rate [16].

For this reason, static energy is very accurately conservative, which explains why it has formed the basis for the study of poleward energy transports in the atmosphere, even though it is not a quasi-material quantity (that is, a function of just  $(\eta, S)$ ). Its boundary fluxes exactly coincide with the enthalpy fluxes both at the surface as well as at the bottom, whereas this is only approximately true for potential enthalpy at the bottom.

**How is $\Sigma_{\text{heat}}$ related to potential enthalpy (McDougall et al. 2021)?**

The APE-based definition of  $\Sigma_{\text{heat}}$  consists in defining it as the value of static energy in Lorenz reference state, that is, as

$$\Sigma_{\text{heat}}(\eta, S) = h(\eta, S, p_r) + \Phi_r \tag{7}$$

where  $p_r = p_0(\Phi_r)$  and  $\Phi_r = \Phi_r(\eta, S)$  are the reference pressure and geopotential of a fluid parcel in Lorenz reference state. Mathematically,  $\Phi_r$  minimises the function  $F(\Phi) = \Sigma(\eta, S, p_0(\Phi)) + \Phi$  at fixed value of  $(\eta, S)$  and is therefore solution of

$$F_0'(\Phi_r) = -\upsilon(\eta, S, p_0(\Phi_r))\rho_0(\Phi_r) + 1 = 0, \tag{8}$$

referred to as the level of neutral buoyancy (LNB) equation in the literature, e.g., [18], in which which  $dp_0/d\Phi = -\rho_0(\Phi)$ . As a result, the evolution equation for  $\Sigma_{\text{heat}}$  can be shown to be

$$\frac{D\Sigma_{\text{heat}}}{Dt} = T_r \frac{D\eta}{Dt} + \mu_r \frac{DS}{Dt}
= -\frac{1}{\rho} \nabla \cdot \mathbf{J}_r + \varepsilon_K + \varepsilon_p,$$
(9)

in which  $T_r = T(\eta, S, p_r)$  and  $\mu_r = \mu(\eta, S, p_r)$ , while  $\varepsilon_p$  represents the APE dissipation, which is generally about 20% of the viscous dissipation rate  $\varepsilon_K$  [16]. As a result,  $\Sigma_{\text{heat}}$  is very accurately conservative.

The same equation is also satisfied by potential enthalpy, except that the latter uses a reference pressure  $p_r = p_a$  corresponding to the mean surface atmospheric pressure.

The boundary fluxes of  $\Sigma_{\text{heat}}$  are equal to the boundary fluxes of enthalpy minus the APE production rate  $G_A$  by surface buoyancy fluxes. According to [4],  $G_A$  rarely exceeds  $30mWm^{-2}$ . The boundary fluxes of  $\Sigma_{\text{heat}}$  are therefore very close to the boundary fluxes of enthalpy. The APE-based  $\Sigma_{\text{heat}}$  therefore corresponds to the definition of heat that most accurately isolate the passive component of potential energy, while being both accurately conservative and having boundary fluxes accurately matching boundary fluxes of enthalpy.

In eq. (5) and below, the partial derivatives should be written in thermodynamic convention indicating which variables are kept constant. Standard mathematical convention is that if one explicitly declares the dependent variables to be  $(\eta, S, p, \Phi)$ , then a derivative such as  $\partial \Sigma/\partial p$  assumes by default that  $(\eta, S, \Phi)$  are held constant. In any case, whether to use mathematical or thermodynamic convention is generally determined by the field of study and personal preferences. I do not know many papers or textbooks in oceanography or atmospheric sciences that still follow thermodynamic convention. Mathematical convention yields simpler formula uncluttered by needless symbols, and is preferred here. I only follow thermodynamic convention when there is a risk of ambiguity or confusion, which is not the case here.

Eqs. (10) and (11) are typical thermodynamic Legendre transforms (Alberty 2002). Presenting exact differentials of  $\Sigma$ ,  $\Sigma_{\rm dyn}$ ,  $E_p$ ,  $E_a$ ,  $E_b$  along with explaining the physical quantities of the implied partial derivatives in terms of standard thermodynamics would greatly help the reader to follow the given arguments.

There is nothing special or surprising in the use of Legendre transform in Eqs. (10) and (11), as it simply corresponds to obtaining internal energy from enthalpy as  $u = h - p\partial h/\partial p = h - p/\rho$ . This being said, I agree that it is useful to clarify how each of the quantities listed can be written in terms of standard thermodynamic quantities. The relevant formula, which I propose to add in an appendix, are:

$$\Sigma = h(\eta, S, p) + \Phi \tag{10}$$

$$\Sigma_{\text{dyn}} = \Sigma - \Sigma_{\text{heat}} = h(\eta, S, p) - h(\eta, S, p_0(\Phi_r)) + \Phi - \Phi_r$$
(11)

$$E_p = \Sigma - p \frac{\partial \Sigma}{\partial p} = h(\eta, S, p) - \frac{p}{\rho} + \Phi$$
 (12)

$$E_a = \Sigma_{\text{dyn}} - p \frac{\partial \Sigma_{\text{dyn}}}{\partial p} = h(\eta, S, p) - h(\eta, S, p_0(\Phi_r)) + \Phi - \Phi_r - \frac{p}{\rho}$$
 (13)

$$E_b = \Sigma_{\text{heat}} - p \frac{\partial \Sigma_{\text{heat}}}{\partial p} = \Sigma_{\text{heat}} = h(\eta, S, p_0(\Phi_r)) + \Phi_r$$
 (14)

Notes:

- The quantity  $E_p$  corresponds to the traditional potential energy, that is, the sum of internal energy  $h p/\rho$  and gravitational potential energy  $\Phi$ ;
- The dynamical component  $E_a$  may be rewritten in the following form

$$E_a = \Pi_1 + \Pi_2 - \frac{p_0(\Phi)}{\rho} \tag{15}$$

where

$$\Pi_1 = h(\eta, S, p) - h(\eta, S, p_0(\Phi)) - \frac{p - p_0(\Phi)}{\rho}$$
(16)

$$\Pi_2 = h(\eta, S, p_0(\Phi)) - h(\eta, S, p_0(\Phi_r)) + \Phi - \Phi_r \tag{17}$$

Physically,  $\Pi_1$  and  $\Pi_2$  are positive definite quantities, generally referred to as available compressible energy (ACE) or available acoustic energy (AAE), and APE density respectively.  $\Pi_1$  represents the compressible work needed to bring the fluid parcel from the reference pressure  $p_0(\Phi)$  to the actual pressure p.  $\Pi_2$  represents the work against buoyancy forces needed to bring a parcel from its reference position  $\Phi_r$  at reference pressure  $p_r = p_0(\Phi_r)$  to the reference pressure  $p_0(\Phi)$  at location  $\Phi$ . These terms have been first defined in local APE theory [19]. Note that the Lagrangian derivative may be written as

$$\rho \frac{DE_a}{Dt} = \rho \frac{D(\Pi_1 + \Pi_2)}{Dt} - \nabla \cdot [p_0(\Phi)\mathbf{v}]$$
 (18)

using the continuity equation  $D\rho/Dt + \rho\nabla \cdot \mathbf{v} = 0$ .

**References**

- [1] D. G. Andrews. A note on potential energy density in a stratified compressible fluid. J. Fluid Mech., 107:227–236, 1981.
- [2] D. Holliday and M. E. McIntyre. On potential energy density in an incompressible, stratified fluid. *J. Fluid Mech.*, 107:221–225, 1981.
- [3] T. G. Shepherd. A unified theory of available potential energy. *Atmosphere-Ocean*, 31:1–26, 1993.
- [4] V. E. Zemskova, B. L. White, and A. Scotti. Available potential energy and the general circulation: partitioning wind, buoyancy forcing, and diapycnal mixing. *J. Phys. Oceanogr.*, 45:1510–1531, 2015.
- [5] L. Novak and R. Tailleux. On the local view of atmospheric available potential energy. J. Atmos. Sci., 75:1891–1907, 2018.
- [6] Z. Liu, C. L. E. Franzke, L. Novak, R. Tailleux, and V. Lembo. A systematic local view of the long-term changes of the atmospheric energy cycle. J. Climate, in press, 2024.
- [7] B. L. Harris, R. Tailleux, C. E. Holloway, and P. L. Vidale. A moist available potential energy budget for an axisymmetric tropical cyclone. *J. Atmos.* Sci., 79:2493–2513, 2022.
- [8] B. L. Harris and R. Tailleux. Diabatic and frictional controls of an axisymmetric vortex using available potential energy theory with a non-resting state. *Atmosphere*, 16(6):700, 2025.
- [9] G. Roullet and P. Klein. Available potential energy diagnosis in a direct numerical simulation of rotating stratified turbulence. *J. Fluid Mech.*, 624:45–55, 2009.
- [10] A. Scotti and B. White. Diagnosing mixing in stratified turbulent flows with a locally defined available potential energy. *J. Fluid Mech.*, 740:114–135, 2014.
- [11] R. Tailleux and G. Roullet. Energetically consistent localised ape budgets for local and regional studies of stratified flow energetics. *Ocean Modelling*, 197:102579, 2025.
- [12] R. Tailleux. Negative available potential energy dissipation as the fundamental criterion for double diffusive instabilities. *J. Fluid Mech.*, 994(A5), 2024.
- [13] J. M. Kempf and F. Rincon. Non-linear saturation and energy transport in global simulations of magneto-thermal turbulence in the stratified intracluster medium. A & A, 694:A25, 2025.

- [14] K. D. Stewart, J. A. Saenz, A. McC. Hogg, G. O. Hughes, and R. W. Griffiths. Effect of topographic barriers on the rates of available potential energy conversion of the oceans. *Ocean Modell.*, 76:31–42, 2014.
- [15] R. Tailleux and T. Dubos. A simple and transparent method for improving the energetics and thermodynamics of seawater approximations: Static Energy Asymptotics (SEA). *Ocean Modelling*, 188(102339), 2024.
- [16] A. F. Waterhouse and al. Global patterns of diapycnal mixing from measurements of the turbulent dissipation rate. J. Phys. Oceanogr., 44:1854–1872, 2014.
- [17] J. M. Gregory and R. Tailleux. Kinetic energy analysis of the response of the atlantic meridional overturning circulation to co2-forced climate change. Clim. Dyn., 37:893–914, 2011.
- [18] R. Tailleux. Available potential energy density for a multicomponent Boussinesq fluid with a nonlinear equation of state. *J. Fluid Mech.*, 735:499–518, 2013.
- [19] R. Tailleux. Local available energetics of multicomponent compressible stratified fluids. *J. Fluid Mech.*, 842(R1), 2018.

---

## Author Comment (AC3)

**Reply to Referee 3**

I thank the referee for their careful reading and thoughtful comments, which help clarify and strengthen the manuscript. I agree that revisions are warranted regarding the observability discussion in Section 3 and the formulation of  $\Sigma_{\rm heat}$  in Section 4. I respectfully disagree, however, that the analogy with gravity in Section 2 is problematic within the stated scope of classical Newtonian mechanics, which is the framework relevant for the oceanographic applications considered.

In what follows, I address each point in turn. I believe the revisions outlined below will resolve the referee's concerns and improve the manuscript.

On the use of gravitational acceleration as an analogy The manuscript treats gravitational acceleration as an example of a field with a global origin that is locally observable. However, this analogy is problematic... [Referee's comment omitted here for brevity in the final manuscript; retained in the editorial correspondence]

**Response** The purpose of the analogy is to contrast the global character of the gravity field—set by the large-scale mass distribution—with the local character of the force entering Newton's second law,

$$\frac{d^2z}{dt^2} = -g(x, y, z, t). \tag{1}$$

This illustrates a duality: a physical quantity can possess both global and local aspects, depending on context. The manuscript argues that the Lorenz reference state (LRS) exhibits a similar duality. Recognising this is important for reframing aspects of APE theory.

I agree that general relativity provides the modern, comprehensive description of gravity. However, the oceanographic setting of this work is Newtonian, where (1) is appropriate and widely used, including in practical determinations of g (e.g., with gravimeters). Within this classical framework, the analogy is scientifically accurate for the intended pedagogical purpose and does not impact the subsequent ocean energetics analysis. I will clarify in the text that the analogy is explicitly Newtonian and is invoked solely to highlight the global–local distinction.

On the observability of the Lorenz reference state (Section 3) Referee: The manuscript suggests that the LRS can, in principle, be reconstructed from locally measured buoyancy frequency if the system is sufficiently close to rest... Consequently, the practical and theoretical usefulness of the arguments presented in Section 3 is unclear.

Response I appreciate the referee's emphasis on realism. Establishing observability in a controlled limit (near-rest conditions) is a necessary first step: if the proposition failed in that limit, it would be unlikely to hold more generally. Demonstrating that the LRS is, in principle, locally inferable from buoyancy

frequency in simple cases provides a clear foundation for arguing that the LRS retains an observable character more broadly, even if the inference becomes more involved in energetic regimes.

To make this explicit, I will revise Section 3 to frame the near-rest analysis as a baseline result and then outline a more general formulation based on anomalous forces. Specifically, let  $p = p_0(z) + \delta p$ , with  $p_0(z)$  dynamically passive. For the non-hydrostatic primitive equations, the momentum balance can be written as

$$\frac{D\mathbf{v}}{Dt} + f\,\mathbf{k} \times \mathbf{u} + \frac{1}{\rho_{\star}} \nabla \delta p = b\,\mathbf{k},\tag{2}$$

where  $b = -g(\rho(S, \theta, p_0(z)) - \rho_0(z))/\rho_{\star}$  and  $\rho_0(z) = -p'_0(z)/g$ . Define the parcel's reference equilibrium level  $z_r = z_r(S, \theta)$  by  $b(S, \theta, z_r) = 0$ , and the displacement  $\zeta = z - z_r$ . For adiabatic and isohaline motion,  $w = Dz/Dt = D\zeta/Dt$ . The vertical momentum balance becomes

$$\frac{D^2\zeta}{Dt^2} + \frac{1}{\rho_{\star}} \frac{\partial \delta p}{\partial z} + \int_0^{\zeta} N_0^2(S, \theta, z_r + \zeta') \, d\zeta' = 0, \tag{3}$$

using

$$b(S, \theta, z) = \int_0^{\zeta} b_z(S, \theta, z_r + \zeta') d\zeta', \tag{4}$$

$$b_z(S, \theta, z) = -N_0^2(S, \theta, z) = \frac{g}{\rho_*} \left( \frac{d\rho_0}{dz}(z) + \frac{g \,\rho_0(z)}{c_s^2(S, \theta, p_0(z))} \right). \tag{5}$$

Equation (3) shows that, for finite-amplitude  $\zeta$ , the vertical motion comprises forced and free nonlinear buoyancy oscillations across a range of processes (turbulence, internal waves, balanced motions), all of which depend—albeit in complex ways—on  $N_0^2$ . Inferring  $\rho_0(z)$  and  $p_0(z)$  in the general case is therefore a nontrivial inverse problem rather than an impossibility. The key point is that the LRS remains tied to observable quantities, preserving its status as an observable construct even away from rest.

This derivation also clarifies an apparent arbitrariness: while  $p_0(z)$  enters as a passive reference, its choice determines  $N_0$  and is thus constrained by observations; there is not an arbitrary family of equally acceptable reference states once observational consistency is imposed. I will incorporate these clarifications in the revised Section 3.

On the formulation using the static energy function (Section 4) Section 4 presents a reformulation in terms of a static energy function  $\Sigma$ ... The author defines  $\Sigma_{\text{heat}}$  based on the LRS... Yet, this statement is questionable, as the choice of  $\Sigma_{\text{heat}}$  in equation (8) is not unique... This arbitrariness weakens the claim that the LRS directly shapes the dynamics...

Response I agree that, formally, one might contemplate alternative definitions of  $\Sigma_{\text{heat}}$ . Physically, however, the distinction between available and non-available energy is meaningful only if the "non-available" part is selected so

that the remaining "available" part exhibits the observed dynamical signatures. This provides a concrete criterion that constrains  $\Sigma_{\text{heat}}$ .

One informative test is the sign and structure of surface-forced production/destruction of  $\Sigma_{\rm dyn}$  by heat and freshwater fluxes. Denoting the net surface heat flux by  $Q_{\rm net}$ , the surface freshwater density by  $\rho_f = \rho(0, T, p)$ , and the net evaporation minus precipitation by E - P (m s-1), we obtain

$$F_{\rm dyn} = \left(\frac{T - T_r}{T}\right) Q_{\rm net} + \left[\mu - \mu_r - (T - T_r) \frac{\partial \mu}{\partial T}\right] \rho_f S(E - P), \tag{6}$$

with

$$T_r = \frac{\partial \Sigma_{\text{heat}}}{\partial \eta}, \qquad \mu_r = \frac{\partial \Sigma_{\text{heat}}}{\partial S}.$$
 (7)

For the APE-consistent choice of  $\Sigma_{\rm heat}$  (i.e., based on the LRS),  $F_{\rm dyn}$  coincides with the exact APE production form [e.g 1, 2, 3] and is positive when surface fluxes destabilise the water column, in agreement with observational and modeling evidence. In contrast, if  $\Sigma_{\rm heat}$  were taken, for example, as potential enthalpy [4], for which  $T_r = \theta$  and  $\mu_r = \mu$ , one obtains  $F_{\rm dyn} = 0$ , implying that surface fluxes do not contribute to APE production—at odds with empirical understanding.

Thus, while multiple mathematical decompositions are possible in principle, the requirement that  $\Sigma_{\rm dyn}$  encode the observed energetics imposes strong physical constraints that single out the LRS-based  $\Sigma_{\rm heat}$  as the relevant choice. I will expand Section 4 to include this argument and additional implications that further reduce any perceived arbitrariness.

**Summary statement** In summary, while the manuscript raises important conceptual questions... I would encourage the author to make it publicly available as a non-refereed contribution...

Response I appreciate the referee's constructive engagement. With the clarifications and additions outlined above—especially the strengthened treatment of observability in Section 3 and the physical constraints on  $\Sigma_{\text{heat}}$  in Section 4—I believe the revised manuscript will address the concerns raised and meet the standards for publication. I will implement these revisions and resubmit.

**References**

- [1] G. O. Hughes, A. M. Hogg, and R. W. Griffiths. Available potential energy and irreversible mixing in the meridional overturning circulation. *J. Phys. Oceanogr.*, 39:3130–3146, 2009.
- [2] J. A. Saenz, R. Tailleux, E. D. Butler, G. O. Hughes, and K. I. C. Oliver. Estimating lorenz's reference state in an ocean with a nonlinear equation of state for seawater. J. Phys. Oceanogr., 45:1242–1257, 2015.

- [3] V. E. Zemskova, B. L. White, and A. Scotti. Available potential energy and the general circulation: partitioning wind, buoyancy forcing, and diapycnal mixing. *J. Phys. Oceanogr.*, 45:1510–1531, 2015.
- [4] T. J. McDougall. Potential enthalpy: a conservative oceanic variable for evaluating heat content and heat fluxes. *J. Phys. Oceanogr.*, 33:945–963, 2003.

---

## Author Comment (AC4)

**Reply to Andy Hogg**

I thank the referee for his thoughtful comments and for his substantial contributions to APE theory over many years, particularly in the global APE framework context. I agree that several issues merit clarification and revision. I address each point in turn below and will implement the indicated changes in the revised manuscript.

Long-range interactions This paper examines the definition of Available Potential Energy as per the Lorenz Reference State (LRS). One significant issue with the LRS is that it is defined relative to the global density field. This means that, in principle, altering the density in the Arctic Ocean may alter the energetics in the Southern Ocean, even though it is unclear how that information could be communicated across the planet.

Response I appreciate the concern. The key point in the manuscript is that the LRS is dynamically passive: it is introduced as a reference against which dynamical anomalies are defined, not as an active agent that transmits influence. As such, whether one regards the LRS as global or local does not introduce new interactions or time/space scales into the dynamics; it is a bookkeeping device that helps uncover dynamical information in the momentum and energy balances that would otherwise remain hidden. Buoyancy oscillations is an example of dynamics that can only be uncovered by the introduction of a dynamically passive reference state.

To make this explicit, I will clarify in Section 3 that the momentum equations are most transparently written in terms of anomalous forces. Writing the pressure as  $p = p_0(z) + \delta p$ , with  $p_0(z)$  dynamically passive, the non-hydrostatic primitive equations can be expressed as

$$\frac{D\mathbf{v}}{Dt} + f\,\mathbf{k} \times \mathbf{u} + \frac{1}{\rho_{\star}} \nabla \delta p = b\,\mathbf{k},\tag{1}$$

where  $b = -\frac{g}{\rho_{\star}} \left( \rho(S,\theta,p_0(z)) - \rho_0(z) \right)$  and  $\rho_0(z) = -p_0'(z)/g$ . The introduction of  $p_0(z)$  does not alter the fundamentally local character of the forces entering the momentum balance, nor does it imply any long-range coupling beyond that already present in the governing equations. I will revise Section 3 to make this point more prominent.

Nature of the Lorenz reference state Since the LRS is often regarded as the zero-APE state, some have interpreted it as "real", yet there is no practical way to attain it in a complex fluid. In most cases it is hypothetical; what is gained by measuring potential energy relative to an unattainable state?

Response This is an important conceptual issue. In the manuscript I propose addressing it in terms of observability. If the LRS is tied to observables (directly or indirectly), then it is not merely a formal or hypothetical construct. This way, its 'reality' can be more objectively assessed and discussed. Two distinct notions are helpful:

- Direct observability: near equilibrium, the LRS relates to locally measurable buoyancy frequencies. Predictions for  $N_0^2$  can be tested against direct measurements of buoyancy oscillation frequencies. Establishing this in a controlled limit is a necessary first step; if the proposition failed near rest, it would be unlikely to hold more generally.
- Indirect observability: different definitions of the "non-available" component (i.e.,  $\Sigma_{\rm heat}$ ) lead to distinct, testable dynamical consequences. For example, the surface-forced production/destruction of the dynamical part  $\Sigma_{\rm dyn}$  depends on the choice of  $\Sigma_{\rm heat}$ . The APE-consistent choice (based on the LRS) yields surface production forms that align with empirical evidence for when surface fluxes destabilize the water column, whereas other choices (e.g., based on potential enthalpy) do not. I will make these distinctions explicit in Sections 3–4 and provide additional examples.

**Local character of the LRS** The paper aims to show that the LRS can be regarded as a local quantity and linked to the governing equations. I would be delighted if this were established, but I remain unconvinced and believe major improvements are needed for publication.

Response Thank you for the candid assessment. The revision will strengthen the argument along two lines: (i) clarifying the controlled, near-rest result as a baseline—a necessary foundation for broader claims—and (ii) setting out a more general anomalous-force formulation that demonstrates how observable quantities (notably  $N_0^2$ ) constrain the reference fields. Together these revisions should render the intent and limitations of the argument clearer.

Limitations of Section 3 The gravity analogy is imperfect; g varies, but can be measured locally. Buoyancy frequency is also measurable locally, which gives a local approximation to APE that suffices for linear internal waves. However, as shown by Hughes, Hogg and Griffiths (2009), the local linear approximation is insufficient for the large-scale overturning and associated mixing. It has not been shown that Section 3's arguments extend beyond the linear range; more work is needed.

**Response** I agree that the local linear approximation has limitations for large-scale energetics and mixing, as highlighted by [1]. Section 3 is not intended to claim sufficiency of the linear approximation for all purposes. Rather, its purpose is to establish a baseline: in the near-rest limit, the LRS couples to local observables such as  $N_0^2$ . This provides a clear "existence proof" for observability in a controlled setting.

To bridge toward more general regimes, I will add the following formulation. Define the parcel's neutral level  $z_r = z_r(S,\theta)$  by  $b(S,\theta,z_r) = 0$  and the displacement  $\zeta = z - z_r$ . For adiabatic, isohaline motion  $w = Dz/Dt = D\zeta/Dt$ , and the vertical balance becomes

$$\frac{D^2 \zeta}{Dt^2} + \frac{1}{\rho_*} \frac{\partial \delta p}{\partial z} + \int_0^{\zeta} N_0^2(S, \theta, z_r + \zeta') \, d\zeta' = 0, \tag{2}$$

with

$$b(S, \theta, z) = \int_0^{\zeta} b_z(S, \theta, z_r + \zeta') d\zeta', \tag{3}$$

$$b_z(S, \theta, z) = -N_0^2(S, \theta, z) = \frac{g}{\rho_*} \left( \frac{d\rho_0}{dz}(z) + \frac{g \rho_0(z)}{c_s^2(S, \theta, p_0(z))} \right). \tag{4}$$

Equation (2) shows that, away from the linear limit, vertical motions comprise forced and free nonlinear buoyancy oscillations across turbulence, internal waves, and balanced motions—each depending (sometimes intricately) on  $N_0^2$ . Inferring  $\rho_0(z)$  and  $p_0(z)$  in these regimes is a nontrivial inverse problem, but not an impossibility. I will clarify these points and add an explicit citation to [1] to acknowledge this limitation and context.

Static energy asymptotics as a basis for APE theory (Section 4) I accept the static-energy framework and that APE can be written from the dynamical component. However, Equation (8) relies on the Lorenz definition of the reference state, which remains global. Thus the separation presumes global knowledge and does not demonstrate an "external constraint." Moreover, there is a genuinely non-local APE effect: e.g., if the northern-hemisphere water were magically made  $10 \, \mathrm{kg} \, \mathrm{m}^{-3}$  denser, it would sink below the southern-hemisphere water, altering energetics remotely.

Response I agree that, formally, different choices of  $\Sigma_{\rm heat}$  are conceivable. The question is whether they are physically acceptable. The decomposition is meaningful only if the "non-available" part is chosen so that the remaining "available" part exhibits the observed dynamical signatures. This imposes strong constraints.

A particularly diagnostic test concerns the surface-forced production or destruction of  $\Sigma_{\rm dyn}$  by heat and freshwater fluxes. Denoting the net surface heat flux by  $Q_{\rm net}$ , the surface freshwater density by  $\rho_f = \rho(0,T,p)$ , and net evaporation minus precipitation by E-P (m s-1), we obtain

$$F_{\rm dyn} = \left(\frac{T - T_r}{T}\right) Q_{\rm net} + \left[\mu - \mu_r - (T - T_r) \frac{\partial \mu}{\partial T}\right] \rho_f S(E - P), \tag{5}$$

with

$$T_r = \frac{\partial \Sigma_{\text{heat}}}{\partial \eta}, \qquad \mu_r = \frac{\partial \Sigma_{\text{heat}}}{\partial S}.$$
 (6)

For the APE-consistent, LRS-based  $\Sigma_{\rm heat}$ ,  $F_{\rm dyn}$  reduces to the exact APE production form [e.g., 1, 2, 3] and is positive when surface fluxes destabilize the water column, consistent with empirical understanding. By contrast, if  $\Sigma_{\rm heat}$  were defined via potential enthalpy [4], for which  $T_r = \theta$  and  $\mu_r = \mu$ , one obtains  $F_{\rm dyn} = 0$ , implying that surface fluxes do not contribute to APE production—contrary to observations and established energetics. I will expand Section 4 with this comparison and additional examples, thereby reducing the perception of arbitrariness and clarifying the "external constraint" language (which I will rephrase to avoid any implication of causal forcing by the LRS itself).

Regarding the thought experiment: an abrupt, global-density modification would take the system far from equilibrium and trigger a global adjustment via acoustic and internal gravity waves and balanced motions. The subsequent reorganisation of APE and BPE reflects the system's dynamical response to the imposed perturbation, not an intrinsic non-local action of APE. I will clarify this distinction in the text.

**Summary** I remain unconvinced that the LRS can be considered a local quantity. The nonlocal generation of APE is less problematic if one avoids treating the LRS as real or attainable. A hierarchy of approximations to the exact LRS (from local-linear to semi-local) is likely the most useful path.

Response I appreciate these suggestions. The revised manuscript will:

- Emphasize observability as the organizing principle (direct in near-rest limits; indirect via dynamical constraints such as surface-forced production).
- Clarify that the LRS's role is as a passive reference constraining the decomposition, not as a driver of dynamics.
- Expand Section 4 to demonstrate how physically consistent constraints single out the LRS-based Σheat.

With these revisions, I believe the manuscript will address the concerns raised and meet the standards for publication.

**References**

- [1] G. O. Hughes, A. M. Hogg, and R. W. Griffiths. Available potential energy and irreversible mixing in the meridional overturning circulation. *J. Phys. Oceanogr.*, 39:3130–3146, 2009.
- [2] J. A. Saenz, R. Tailleux, E. D. Butler, G. O. Hughes, and K. I. C. Oliver. Estimating lorenz's reference state in an ocean with a nonlinear equation of state for seawater. *J. Phys. Oceanogr.*, 45:1242–1257, 2015.
- [3] V. E. Zemskova, B. L. White, and A. Scotti. Available potential energy and the general circulation: partitioning wind, buoyancy forcing, and diapycnal mixing. *J. Phys. Oceanogr.*, 45:1510–1531, 2015.
- [4] T. J. McDougall. Potential enthalpy: a conservative oceanic variable for evaluating heat content and heat fluxes. *J. Phys. Oceanogr.*, 33:945–963, 2003.

---

## Author Comment (AC5)

**Reply to Referee 3**

**Referee** I appreciate the author's efforts in addressing the reviewers' comments. However, after carefully evaluating the responses, I find that several major concerns remain insufficiently resolved.

Response I thank the referee for their additional comments and for engaging with the revised arguments. The local APE framework is still at an early stage of development. Over the past decade I have worked extensively on this topic, with 16 publications on local APE and closely related issues in both the ocean and the atmosphere [1, 2, 3, 4, 5, 6, 7, 8, 9, 10, 11, 12, 13, 14, 15, 16]. Inevitably, this means that many colleagues who kindly agree to review my work bring a broad physical perspective rather than detailed familiarity with this specific theory.

In what follows, I will clarify where I believe the referee's concerns arise from differing interpretations, and I will strengthen the manuscript where the current presentation may have contributed to such differences. My aim is to make the underlying assumptions and logical steps as explicit as possible so that readers can fairly assess the framework and its implications.

Referee: On the use of gravitational acceleration as an analogy Even when restricting the discussion to Newtonian mechanics, gravity does not seem to constitute an appropriate analogy for the Lorenz reference state (LRS) ... Determining the LRS corresponding to a given spatial distribution of S and  $\theta$  requires a global rearrangement of fluid parcels, which is inherently difficult to express in a local form. This is precisely why the physical interpretation of the LRS has remained a challenging issue.

Response I agree that the LRS is more intricate than the gravitational potential field, and that its physical interpretation has long been non-trivial. The analogy with gravity is not meant to suggest that the LRS can be written as a simple Poisson problem, but to highlight a conceptual parallel: both are fields whose local values are constrained by global distributions (mass for gravity, water-mass properties for the LRS) and yet are used locally in the dynamics.

In fact, there exist explicit formulations that link the reference density profile to the water-mass distribution in a way that is directly analogous, in spirit, to the Poisson equation. For a Boussinesq fluid in a simple vertical-walled basin, [17] derived the following explicit expression for the reference position of a parcel,  $z_{\star}(\mathbf{x},t)$ :

$$z_{\star}(\mathbf{x},t) = \frac{1}{A} \int H(\rho(\mathbf{x}',t) - \rho(\mathbf{x},t)) dV', \qquad (1)$$

where H is the Heaviside function and A is the (constant) horizontal area; see their Eq. (11). This can be inverted to yield  $\rho_0(z,t)$ .

For a realistic ocean with nonlinear equation of state and variable basin geometry, [12] generalised this construction. Their approach defines  $z_r(\rho)$ , the

inverse of  $\rho_0(z)$  such that  $\rho_0(z_r(\rho)) = \rho$ , as the solution of

$$\int_{z_r(\rho)}^{0} A(z) dz = V_T \int_{\theta_{\min}}^{\theta_{\max}} \int_{\hat{S}(\theta, \rho, p_0(z_r(\rho)))}^{\hat{S}(\theta, \rho_{\min}, p_0(0))} \Pi(S, \theta) dS d\theta,$$
 (2)

where A(z) is the ocean area at depth z,  $V_T = \int_{-H_{\text{max}}}^{0} A(z) dz$  is the total volume, and  $\Pi(S, \theta)$  is the normalised volume distribution function (see their Eq. (29) and discussion).

In both cases, the determination of the reference density profile can be written abstractly as

$$\mathcal{L}(z_r(\rho)) = 0, \tag{3}$$

where  $\mathcal{L}$  is an operator that explicitly links the local value of  $z_r(\rho)$  to the global water-mass distribution. While (3) is not identical to the Poisson equation (??), the structure is similar in that a local reference field is determined by an integral or differential relation involving the global state.

I will revise the manuscript to: - Cite [17] and [12] explicitly and explain how their constructions provide a mathematically well-defined route from the global distribution to a locally defined reference profile. - Clarify that the gravity analogy is conceptual and pedagogical: it aims to illustrate a dual "global-local" character, not to suggest an exact mapping of operators.

Referee: On the observability of the Lorenz reference state The theoretical argument presented through equations (2)–(5) is not convincing. While the mathematical manipulations themselves are correct, the reference profiles  $\rho_0(z)$  and  $p_0(z)$  introduced here may be chosen arbitrarily; there is no inherent reason to identify them with the LRS. Furthermore, the statement that "while  $p_0(z)$  enters as a passive reference, its choice determines  $N_0$  and is thus constrained by observations" is not meaningful. The authors may be implicitly assuming that  $N_0$  corresponds to the in-situ Brunt-Väisälä frequency, but this is not the case . . . In contrast, the quantity  $N_0$  defined in the author's reply depends solely on the arbitrarily chosen reference state and is therefore not observable.

Response The exact vertical momentum balance,

$$\frac{D^2 \zeta}{Dt^2} + \frac{1}{\rho_{\star}} \frac{\partial \delta p}{\partial z} + \int_0^{\zeta} N_0^2 \left( S, \theta, z_r + \zeta' \right) d\zeta' = 0, \tag{4}$$

and its small-amplitude approximation,

$$\frac{D^2 \zeta}{Dt^2} + N_0^2(S, \theta, z_r) \zeta \approx 0, \tag{5}$$

provide a first-principles derivation of the buoyancy frequency in terms of displacements about a neutral reference level  $z_r(S,\theta)$  (defined by  $b(S,\theta,z_r)=0$ ). This construction is deductive and rests only on the governing equations and the definition of neutral buoyancy.

To relate  $N_0^2$  to observed salinity and temperature profiles, the classical environmental approach sets

$$\rho_0(z) = \rho(S_0(z), \theta_0(z), p_0(z)),$$

with  $S_0(z)$  and  $\theta_0(z)$  describing locally defined environmental profiles. Under the assumption that  $\rho_0(z)$  is obtained from the actual state by adiabatic and isohaline rearrangement, and that  $S = S_0(z_r)$ ,  $\theta = \theta_0(z_r)$ , one recovers

$$N_0^2(S, \theta, z_r) = g\left(\alpha \frac{\partial \theta_0}{\partial z}(z_r) - \beta \frac{\partial S_0}{\partial z}(z_r)\right),\tag{6}$$

which is the familiar expression for the (environmental) squared buoyancy frequency used in oceanography. In this setting:

- $N_0^2$  is not arbitrary; it is fixed by the choice of  $\rho_0(z)$ .
- $\rho_0(z)$  is itself constrained by the requirement that it correspond to the Lorenz reference state obtained by adiabatic, isohaline rearrangement.
- The observable quantities are the buoyancy oscillation frequencies (from, e.g., internal waves) and the environmental gradients  $(\partial_z S_0, \partial_z \theta_0)$ ; these jointly constrain the admissible  $\rho_0(z)$  and hence the LRS.

In contrast, the quantity introduced by the referee,

$$N_{\text{inst}}^2 = -\frac{g}{\rho_{\star}} \left( \rho_{\theta}(S, \theta, p_0(z)) \frac{\partial \theta}{\partial z} + \rho_S(S, \theta, p_0(z)) \frac{\partial S}{\partial z} \right), \tag{7}$$

is based on instantaneous vertical gradients and therefore differs conceptually from the environmental buoyancy frequency in (6). In a turbulent ocean,  $N_{\rm inst}^2$  can be highly variable and may take positive or negative values locally, reflecting transient overturning and small-scale variability, and represents a fundamentally meaningless approach to defining the buoyancy frequency. By contrast,  $N_0^2$  derived from  $\rho_0(z)$  represents a smoothed, underlying reference stratification associated with the LRS.

In the manuscript, "observability" is used in the sense of determining  $\rho_0(z)$  (and thus the LRS) from measurements of buoyancy oscillations (direct observability) and from the consistency of various dynamical constraints (indirect observability). I will revise Section 3 to:

- Clearly distinguish between instantaneous, small-scale estimates of  $N^2$  and the reference  $N_0^2$  associated with  $\rho_0(z)$ .
- Emphasise that  $\rho_0(z)$  is not arbitrary: it is selected by the Lorenz rearrangement and constrained by the requirement that the resulting  $N_0^2$  be consistent with observed buoyancy behaviour and environmental structure.
- Clarify the precise sense in which  $N_0^2$  is "constrained by observations" and how this links to the LRS.

Referee: On the formulation using the static energy function  $As\ I$  noted in my previous comments, the decomposition  $\Sigma = \Sigma_{\rm heat} + \Sigma_{\rm dyn}$  and the use of the LRS to define the latter is interesting. However, I remain unconvinced by the strengthened arguments. The author's example involving surface diabatic processes shows that choosing  $\Sigma_{\rm heat}$  based on the LRS makes expression (6) represent the APE production rate, but this is not unexpected since APE is defined relative to the LRS. It reiterates a well-known result and does not substantively support the claim that "the Lorenz reference state enters the equations in the way an external constraint would."

Response I agree that, at first sight, it may appear unsurprising that a decomposition built around the LRS recovers standard APE production forms. The purpose of the example, however, is to show that physically meaningful behaviour is obtained only for a very restricted class of choices for  $\Sigma_{\rm heat}$ , and that this class is tied to the LRS. In other words, the decomposition is not arbitrary.

To illustrate this, consider the surface-forced production/destruction of  $\Sigma_{\rm dyn}$  by heat and freshwater fluxes. Denoting by  $Q_{\rm net}$  the net surface heat flux, by  $\rho_f = \rho(0,T,p)$  the surface freshwater density, and by E-P the net evaporation minus precipitation (m s-1), one finds

$$F_{\rm dyn} = \left(\frac{T - T_r}{T}\right) Q_{\rm net} + \left[\mu - \mu_r - (T - T_r) \frac{\partial \mu}{\partial T}\right] \rho_f S(E - P), \quad (8)$$

with

$$T_r = \frac{\partial \Sigma_{\text{heat}}}{\partial \eta}, \qquad \mu_r = \frac{\partial \Sigma_{\text{heat}}}{\partial S}.$$
 (9)

For the LRS-based, APE-consistent choice of  $\Sigma_{\rm heat}$ ,  $F_{\rm dyn}$  reduces to the exact APE production form [e.g., 18, 12, 19] and is positive when surface fluxes destabilise the stratification, in line with empirical evidence and established energetics.

By contrast, if  $\Sigma_{\text{heat}}$  is defined in terms of potential enthalpy [20], for which  $T_r = \theta$  and  $\mu_r = \mu$ , one obtains  $F_{\text{dyn}} = 0$ , implying that surface fluxes do not contribute to APE production. This is at odds with both observations and standard theoretical understanding. This simple comparison demonstrates that:

- The choice of  $\Sigma_{\rm heat}$  is strongly constrained by dynamical consistency; not all mathematically admissible decompositions yield physically acceptable behaviour. - The LRS-based definition of  $\Sigma_{\rm heat}$  is singled out by these constraints, supporting the view that the LRS plays a distinguished role in the energetic decomposition.

Regarding the phrase "enters the equations in the way an external constraint would": I will soften and clarify this wording in the manuscript to avoid any implication that the LRS exerts a causal forcing. The intended meaning is that once  $\Sigma_{\rm heat}$  is fixed by the LRS, the form of  $\Sigma_{\rm dyn}$  and of the associated source terms (such as  $F_{\rm dyn}$ ) is determined and imposes non-trivial constraints on the dynamics—much as an externally imposed constraint restricts the set of admissible states. I will rephrase this to emphasise the constraining role, rather than suggesting an additional external agent.

In summary, while I recognise that some of these issues involve subtle conceptual distinctions, I believe that with the clarifications and additions outlined above—particularly concerning (i) the mathematical constructions underpinning the LRS, (ii) the precise notion of observability used, and (iii) the physical constraints on  $\Sigma_{\rm heat}$ —the revised manuscript will address the remaining concerns and present a coherent and testable framework for local APE theory.

**References**

- [1] R. Tailleux and G. Roullet. Energetically consistent localised ape budgets for local and regional studies of stratified flow energetics. *Ocean Modelling*, 197:102579, 2025.
- [2] B. L. Harris and R. Tailleux. Diabatic and frictional controls of an axisymmetric vortex using available potential energy theory with a non-resting state. *Atmosphere*, 16(6):700, 2025.
- [3] Z. Liu, C. L. E. Franzke, L. Novak, R. Tailleux, and V. Lembo. A systematic local view of the long-term changes of the atmospheric energy cycle. *J. Climate*, in press, 2024.
- [4] R. Tailleux. Negative available potential energy dissipation as the fundamental criterion for double diffusive instabilities. *J. Fluid Mech.*, 994(A5), 2024.
- [5] R. Tailleux and T. Dubos. A simple and transparent method for improving the energetics and thermodynamics of seawater approximations: Static Energy Asymptotics (SEA). Ocean Modelling, 188(102339), 2024.
- [6] B. L. Harris, R. Tailleux, C. E. Holloway, and P. L. Vidale. A moist available potential energy budget for an axisymmetric tropical cyclone. *J. Atmos.* Sci., 79:2493–2513, 2022.
- [7] B. L. Harris and R. Tailleux. Assessment of algorithms for computing moist available potential energy. Q. J. Roy. Met. Soc., 144:1501–1510, 2018.
- [8] L. Novak and R. Tailleux. On the local view of atmospheric available potential energy. *J. Atmos. Sci.*, 75:1891–1907, 2018.
- [9] R. Tailleux. Local available energetics of multicomponent compressible stratified fluids. J. Fluid Mech., 842(R1), 2018.
- [10] R. Tailleux. Generalized patched potential density and thermodynamic neutral density: Two new physically based quasi-neutral density variables for ocean water masses analyses and circulation studies. J. Phys. Oceanogr., 46:3571–3584, 2016.

- [11] K. C. Wong, R. Tailleux, and S. L. Gray. The computation of reference state and ape production by diabatic processes in an idealized tropical cyclone. Q. J. R. Meteorol. Soc., 142:2646–2657, 2016.
- [12] J. A. Saenz, R. Tailleux, E. D. Butler, G. O. Hughes, and K. I. C. Oliver. Estimating lorenz's reference state in an ocean with a nonlinear equation of state for seawater. *J. Phys. Oceanogr.*, 45:1242–1257, 2015.
- [13] R. Tailleux. Available potential energy density for a multicomponent Boussinesq fluid with a nonlinear equation of state. *J. Fluid Mech.*, 735:499–518, 2013.
- [14] R. Tailleux. Available potential energy and exergy in stratified fluids. *Ann. Rev. Fluid Mech.*, 45:35–58, 2013.
- [15] E. D. Butler, K. I. C. Oliver, J. M. Gregory, and R. Tailleux. The ocean's gravitational potential energy budget in a coupled climate model. *Geophys. Res. Lett.*, 40:5417–5422, 2013.
- [16] R. Tailleux. Irreversible compressible work and available potential energy dissipation. *Phys. Scripta*, T155:014033, 2013.
- [17] K. B. Winters, P. N. Lombard, J. J. Riley, and E. A. d'Asaro. Available potential energy and mixing in density stratified fluids. *J. Fluid Mech.*, 289:115–128, 1995.
- [18] G. O. Hughes, A. M. Hogg, and R. W. Griffiths. Available potential energy and irreversible mixing in the meridional overturning circulation. *J. Phys. Oceanogr.*, 39:3130–3146, 2009.
- [19] V. E. Zemskova, B. L. White, and A. Scotti. Available potential energy and the general circulation: partitioning wind, buoyancy forcing, and diapycnal mixing. *J. Phys. Oceanogr.*, 45:1510–1531, 2015.
- [20] T. J. McDougall. Potential enthalpy: a conservative oceanic variable for evaluating heat content and heat fluxes. *J. Phys. Oceanogr.*, 33:945–963, 2003.